# MoEP: Compact and Efficient Sparsity with Modular Expert Paths

## Abstract

The transition from dense model architectures to sparse ones has become a key trend in the field of Large Language Models (LLMs). Using methods like Mixture-of-Experts (MoE) allows language models to scale their representation power without overloading computation, by relying on sparse parameter activation. Despite this more lightweight activation, the standard MoE approach increases the total number of parameters. This trade-off between size and sparsity can be avoided without losing performance compared to a dense baseline architecture. We introduce MoEP (Modular Expert Paths) as a solution to add sparsity while keeping the total parameter count fixed. MoEP combines model parallelism with MoE-style linear projections to implement selective token activation, which accelerates model learning and enables it to outperform the GPT-2 baseline. This opens a promising research direction, where compact models can still benefit from sparsity.

## 1 Introduction

Looking at the recent landscape of open Large Language Models (LLMs) reveals a clear trend emerging among the best performing models exemplified in (OpenAI et al. (2025), Meta-AI (2025) [1], DeepSeek-AI et al. (2025) and Jiang et al. (2024)). Instead of relying on dense parameter activation, where every token passes through the same route, these models benefit from a scaled representation power enabled by Mixture-of-Experts (MoE). To avoid excessive computational overhead, MoE is typically implemented in a sparse manner, where only a subset of model parameters is activated at the token level for each forward pass. Although sparsity enables the scaling of the representation with agility, it also requires managing a substantially larger total parameter count.

Recent and previous work have examined **sparse and routing-based models** (DeepSeek-AI et al., 2024; Muennighoff et al., 2025; Du et al., 2022; Jiang et al., 2024; OpenAI et al., 2025) and

Recent works have also examined alternative sparsity approaches to **enhance efficiency** and **rethinking sparsity** novel way. Mixture of Recursions (Mor) (Bae et al., 2025) uses depth based sparsity, where different tokens communicates with different depth stack of layers to achieve efficiency. While compositional approach *PaPaformer* (Tapaninaho & Oussala, 2025), introduced method of remodeling Transformer layers into smaller **parallel sub-paths**, which can be used as independently trainable modules.

These efforts highlight a broader trend to improve efficiency and flexibility by enabling tokens to follow different computation paths.

This paper presents *MoEP* (Modular Expert Paths), which adds model sparsity by unifying two forms of routing within a decoder-only language model: (i) **Top-k** token routing across parallel Transformer blocks, and (ii) **Mixture-of-Experts** feed-forward layers based on lightweight linear projections and **SwiGLU** variants. As a result, each token activates only a limited set of parallel blocks and experts in forward-pass, creating more diverse computational pathways while reducing redundancy. In the training phase, a load-balanced **auxiliary loss** was used to encourage the use of stable expert and block utilization without collapse.

---

[1] At the time of writing this paper, no peer-reviewed publication exists for Llama 4, which is why we cite the official announcement. More background and detailed information on the Llama family of models can be found in Touvron et al. (2023a) and Touvron et al. (2023b)

We train our MoEP and MoEP-SwiGLU models with the **BabyLM** strict-small track [2] data and used their official evaluation pipeline.

Despite of GPT-2 outdatedness, we use it as a layer-wise baseline architecture for MoEP to align with the BabyLM GPT-2 baseline model and to keep designs simple. However, MoEP is not architecture bounded and same ideology can be used as well more recent dense architectures to enabling them to employ sparsity. In addition to GPT-2, BabyLM provided GPT-BERT (Charpentier & Samuel, 2024) as alternative baseline, which combines causal and masked language modeling into masked next-token prediction.

Under the official evaluation, MoEP was able to outperform all BabyLM strict-small baseline models, including the GPT-2 and GPT-BERT models as well.

We also show that improving routing mechanism,increased performance within parallel architecture even thought MoEP did not employ the *PaPaformer* (Tapaninaho & Oussala, 2025) style of modularity, in which independent modules are partly pre-trained separately, which prior work suggest as the major of the performance increase.

The main contributions of this work are summarized below:

1. We propose *MoEP*, a modular sparse decoder-only architecture that integrates layer level expert networking across parallel blocks.

2. We provide a BabyLM-compliant evaluation on the strict-small track, comparing MoEP against GPT-2 and other baseline models under matched conditions.

3. We analyze expert networks routing behavior and show that layer level parallelism enable fast and stable training.

4. We introduce a SwiGLU-based MoEP variant, which show that sometimes lightweight simplicity is better than adding complexity.

## 2 BACKGROUND

### 2.1 PARALLEL ARCHITECTURES

As an alternative to dense and sparse expert styles of architecture design, some works have demonstrated interest in parallelisation to increase expressiveness or efficiency. Despite its promising results, this trend has received little attention in the LLM research community. PaLM (Chowdhery et al., 2022) uses parallelisation within layers to achieve faster training with only minor quality degration, while Branchformer (Peng et al., 2022) combines attention and multi-layer-perceptron (MLP) with convolutional gating into parallel components to capture both local and global context. More recently, Papaformer (Tapaninaho & Oussala, 2025) proposed an alternative approach where independently trained parallel paths are combined into a larger composite model. The aim of this work is to achieve efficiency by scaling down dimensionality across parallel layers, while maintaining performance through the increased representation enabled by parallelism. MoEP builds on this idea by maintaining parallelism but coupling it with sparse MoE-style top-k routing.

### 2.2 MIXTURE-OF-EXPERTS

#### 2.2.1 DENSE VS. SPARSE EXPERT ARCHITECTURES

After MoE adaptation into Transfomer-based language models (Vaswani et al., 2023), sparse expert architectures have become almost the standard way to increase representation power without adding density. This most often means that the model is directly pre-trained using a sparse network of experts in some part of the architecture. However, there are several works where the pre-trained model itself is dense, but in the fine-tuning phase sparsity is added using methods like Low-Rank Adaptation (LoRa) (Hu et al., 2021).

These models benefit from stability in the pre-training phase, where all model parameters are activated and tuned during each step, while in the fine-tuning phase switching to sparsity enables

---

[2]https://babylm.github.io

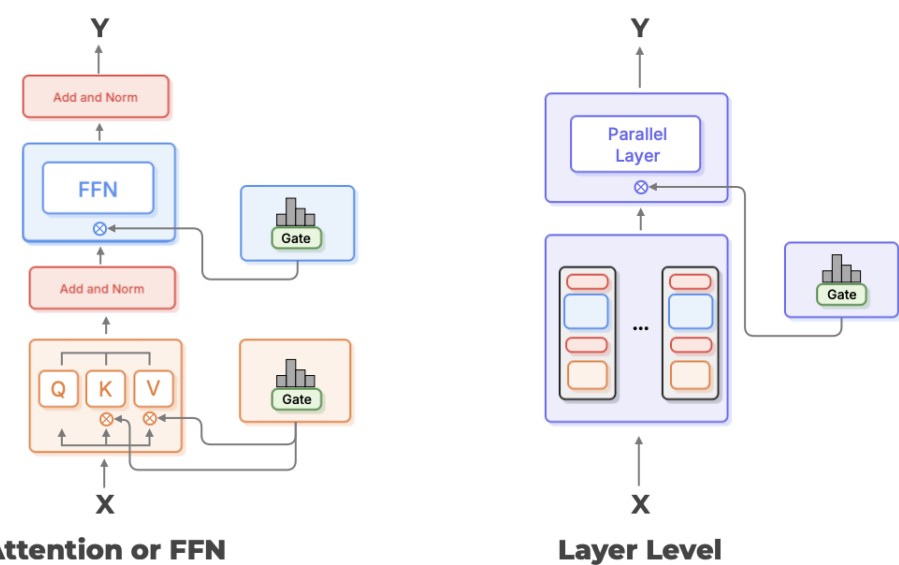

Figure 1: Expert networks placement strategies in Transformer layer. **Left:** sublayer-level expert networks, which routes inputs within Attention, FFN or both. **Right:** ourEx layer-level MoEP, which Parallel layer router selects top-$k$ experts among $P$ parallel blocks (identical structure, distinct parameters), and aggregates their outputs.

stronger specialization and increased representation power. While the most common approach among models that are sparsely pre-trained is to replace the FeedForward network with a network of experts, there is much more variation in sparsity strategies among models that add sparsity after dense pre-training. For example, in fine-tuning phase, MoLe (Wu et al., 2024) uses LoRA weights at the layer level for a pre-trained and frozen model, while LoraMoE (Dou et al., 2024) instead of applies LoRA weights only at the FFN level. DS-MoE Pan et al. (2024) takes a different approach, where expert networks are trained as dense but, during inference, the model switches to sparse activation to achieve efficiency.

Among sparse expert architectures, there are also some less typical approaches, such as THOR (Zuo et al., 2022), which randomly selects experts during training, and MoE-ECR (Zhou et al., 2022), where the FFNs selects subset of tokens instead of that token itself make selection between FFN experts.

### 2.2.2 MoE Placement within Transformer Architecture

#### FFN-Level MoE

Since FeedForward networks dominated the parameter count in Transformer models, most common approach is to replace the feed-forward sublayer with expert networks. Many previous works have explored different methods to optimize efficiency and improve performance. Switch Transformers Fedus et al. (2022) use top-1 routing among experts and achieve significant increases in pre-training speed. LoRAMoE (Dou et al., 2024) freezes pre-trained model weights and use LoRA to create expert networks, while AdaMix (Wang et al., 2022) uses adapter-based expert networks after the FNN. Statte-of-the-art-models such as DeepSeek-R1 DeepSeek-AI et al. (2025) and Llama (Meta-AI, 2025) employ both routed and shared experst, whereas GPT-OSS OpenAI et al. (2025) and Mistral Jiang et al. (2024) advocate routed experts.

#### Attention-Level MoE

Typically attention level experts operate on key and value projections, enabling specialized attention heads that capture diverse relational patterns. This results as a tradeoff an increase in routing com-

plexity inside attention mechanism in comparison to FFN-level routing. Most of the works which used this approach is related to parameter-efficient-finetuning (PEFT). SiRA Zhu et al. (2023) uses LoRA for query and key weights in attention, while pre-trained model is kept freezen. MoA Zhang et al. (2022) on the other hand does not use trainable weights, instead, it uses heterogeneous elastic rules for sparse pattern selection.

### ATTENTION + FFN

Models such as JetMoE Shen et al. (2024) use expert network concurrently at both Attention and FFN levels and outperform Llama and DeepSeek model of comparable size. Mov Zadouri et al. (2023) introduces an interesting approach, where instead of using LoRA directly to model weights, it employs a mixture-of-vectors that are added to the intermediate results in Attention and FFN sublayers.

### LAYER-LEVEL MOE

Instead of replacing the Attention or FFN sublayer, layer-level MoE uses parallel layers as experts as illustrated in Figure 1. This approach provides flexible and sparse pathway selection at the block level. Layer-level expert networks remain a relatively unexplored area, with the exception of MoLE (Wu et al., 2024), which applies LoRa weights at the layer level to a frozen pre-trained model.

The intention of this work is to address these gaps with a novel approach, where parallel expert layers operate at smaller dimensionality. In this way, the overall parameter size is kept fixed while model sparsity is increased.

## 3 METHODOLOGY

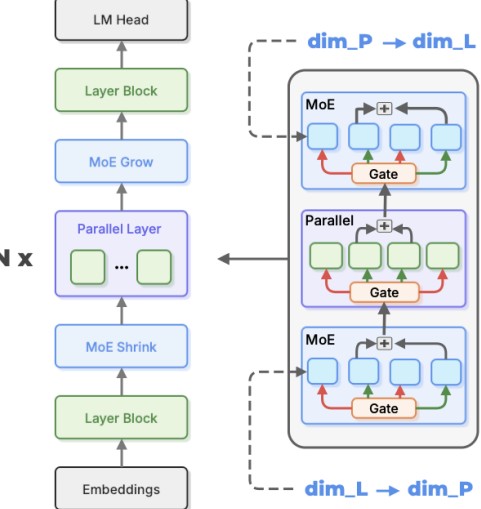

Figure 2: MoEP architecture visualization. $N$ parallel layers are stacked before and after the MoE blocks, whose task is to reduce or increase the hidden dimension to match the layer blocks. In a Parallel layer, the Layer blocks operate on a smaller hidden dimension compared to the individual Layer blocks at the beginning and end of the model.

### 3.1 OVERVIEW OF THE MOEP BASELINE ARCHITECTURE

In this work, the MoEP baseline architecture (see Figure 2) interleaves two standard GPT-2 (dense) layers with a sparse middle stack: **Layer** (full size) → **MoE Block** (shrink) → **Parallel Layer** (repeated N times with top-k gating) → **MoE Block** (grow) → **Layer** (full size).

Both the initial and terminal **Layer** operate at a higher hidden dimension $d_L$. The first **MoE Block** uses $E$ experts with top-$k$ gating to map projection into smaller hidden dimension $d_P$ suited for the parallel stack (see Figure 2). Intermediate **Parallel Layer** uses top-$k$ gating among $P$ **Layer**, which operates at a corresponding smaller hidden dimension $d_P$. After the $N$ Parallel Layer, the second **MoE Block** performs the inverse procedure, mapping the projection from $d_P$ back to $d_L$ before the terminal **Layer**.

### 3.2 MoE Block Projections (Shrink and Grow)

Alternating between two hidden dimensionality $d_L$ and $d_P$, there are corresponding **MoE Block**s to ensure a smooth transition and avoid a possible information bottleneck, where overly rapid dimensionality changes could potentially discard useful information.

The two **MoE Block** projections implement the following dimensionality transition:

$$d_L \; \rightarrow \; \text{shrink} \quad d_P \; \rightarrow \; \text{grow} \; d_L \tag{1}$$

Both **MoE Blocks** consist of $E$ experts with token-level top-k gating over experts. In our **MoEP** baseline model, experts are simple *linear* projections and in **MoEP-SwiGLU**, experts use *SwiGLU*-based feed-forward projections.

### 3.3 Parallel Layers

Each **Parallel Layer** contains $P$ **Parallel blocks** $\{B_1, \ldots, B_K\}$, which are architecturally equivalent to the **Layer Block**, but operate at a reduced dimension $d_P$. The Parallel Blocks share indentical sublayer structure but have fully disjointed parameters. Linear router is shaped $d_P$ x $P$ and it applies a token-level **top-$k$** selection among the $P$ **Parallel Block**, where the routed inputs are summed up together. This routing method implements a layer level expert networking, allowing different tokens to traverse different subsets of Parallel blocks (e.q. layers) within each Parallel Layer. Stacking $N$ **Parallel Layers** yields a deep routed path in compact dimensions.

### 3.4 Routing Objective and Training Loss

To avoid expert (MoE) and Parallel block (Parallel Layer) collapse, which is a typical problem in sparse expert networks, we use the standard load-balancing regularizer in the training phase . More specifically, let $p_i$ denote the average routing probability assigned to Parallel block or expert $i$ over a batch. The balancing term is defined:

$$\mathcal{L}_{\text{balance}} = -\sum_i p_i \log p_i \tag{2}$$

computed separately for Parallel block routing and expert routing. The total objective is then:

$$\mathcal{L} = \mathcal{L}_{\text{CE}} + \lambda^{\text{block}} \mathcal{L}_{\text{balance}}^{\text{block}} + \lambda^{\text{expert}} \mathcal{L}_{\text{balance}}^{\text{expert}} \tag{3}$$

where $\mathcal{L}_{\text{CE}}$ is the next-token cross-entropy loss and $\lambda$ learning weight.

## 4 Experimental Setup

### Training Pipeline

To explore our idea with low resource setup, we follow **BabyLM** (Charpentier et al., 2025) training and evaluation-pipeline in addition to further training analysis. Initially we trained **GPT-2** style **byte-pair encoding** (BPE) tokenizer with fixed vocabulary of 16K and similar pattern-recognition strategy as in [3], ensuring maximal similarity with the **BabyLM** baseline models. Both tokenizer and

---

[3]`https://huggingface.co/BabyLM-community/babylm-baseline-10m-gpt2`

all models were trained by using BabyLM strict-small training data, which contains a little over 10 million words from curated English sources.

### TRAINING PROCEDURE

We trained **MoEP**, **MoEP-SwiGLU** to explore layer level sparsity and **GPT-2** to ensure matching overall performance with **BabyLM** GPT-2 baseline model. To achieve stable model training, we used textbfAdamW with **cosine learning rate decay** with the standard **dropout** and **weight decay** regularization. We initially pre-tokenized the training data with a **stride of 128** and during training, examples were randomly sampled from the full pre-tokenized dataset using an epoch-based shared seed, ensuring that all models were trained on the same examples. This pre-training process followed **BabyLM** strict-small instructions, where each model was trained for 10 epoch and each epoch were stopped after the model had seen approximately 10M words.

During the pre-training, checkpoints were saved every 1M words up to 9M words, and subsequently every 10M words up to 100M words. After this, we ran fast evaluation on all checkpoints, and the final model weights, which was used on full evaluation and further analysis, were taken from the checkpoint with the best evaluation performance. This procedure show notable differences between model learning efficiency. **MoEP** and **GPT-2** achieved their best accuracy at 30M words, while **MoEP-SwiGLU** reached its peak until after 80M words. A detailed table of model based hyperparameters (hidden dimension, number of layers, parameter counts) is listed in the Appendix A.1 and Appendix A.2.

### EVALUATION PROTOCOL

Our model evaluation followed the official BabyLM pipeline (Charpentier et al., 2025), which contained both Zero-shot and Fine-tuning tasks. Zero-shot evaluation included **BLiMP**, **EWOK**, **WUG**, and other tasks, with the full list available in the evaluation pipeline documentation[4]. For tasks like **MNLI**, **QQP**, **RTE**), which involves fine-tuning, the **BabyLM** evaluation pipeline supplied both training data and default finetuning parameters that we adopted directly.

### IMPLEMENTATION ENVIRONMENT

We conducted all our experiments using a single NVIDIA A100 GPU in CSC's Puhti supercomputing environment (CSC – IT Center for Science). Training a single model with 10 epochs required approximately 1-2 hours. This duration could be further reduced with code optimizations. All the model code is implemented using **PyTorch** and **Hugging Face** libraries and released for reproducibility [5] and model is directly downloadable in Hugging Face [6].

## 5 ANALYSIS

### 5.1 EVALUATION SCORES

MoEP achieved the highest performance across all models, including the official **BabyLM** baselines under the strict-small track, when the **AoA** task score was included in the **Macro Average**. Even when excluding AoA from the macro average, MoEP still outperformed the **BabyLM** GPT-2 baseline, which we consider our primary comparison point due to the corresponding sublayer structure. MoEP also obtained the best score in five individual tasks, the highest count among all models evaluated.

Our GPT-2 version slightly outperformed the **BabyLM GPT-2 baseline** in macro average excluding **AoA**, reaching performance near comparable to **MoEP**.

However, our analysis revealed a key distinction (see Appendix A.3), that **MoEP** extracted useful patterns earlier during training, which indicates that modular sparse routing can provide better sample efficiency, even if final scores converge to similar levels.

---

[4]`https://github.com/babylm/evaluation-pipeline-2025/`
[5]`https://github.com`
[6]`https://huggingface.co`

| Model | Zero-shot Tasks | | | | | | | Finetuned Tasks | | | | | | | Macro |
|---|---|---|---|---|---|---|---|---|---|---|---|---|---|---|---|
| | BLiMP | EWOK | Entity | WUG | Comps | Reading | AoA | BoolQ | MNLI | MRPC | MultiRC | QQP | RTE | WSC | Avg |
| *Our Models* | | | | | | | | | | | | | | | |
| GPT-2 | 59.70 | **57.85** | 13.15 | 36.00 | 51.20 | 6.40 | – | 67.50 | 49.10 | 69.60 | 66.70 | 71.55 | **62.60** | 63.45 | 48.10 
 – |
| MoEP [7] | 59.15 | 50.20 | **35.65** | 33.00 | 50.70 | **6.70** | 53.70 | 66.20 | 48.10 | 70.10 | 64.50 | 70.75 | **62.60** | **67.30** | 49.00 
 **44.50** |
| MoEP (SwiGLU) [8] | 60.35 | 49.50 | 17.10 | 36.50 | 51.35 | 6.60 | – | 66.30 | 48.30 | 70.60 | 67.25 | 69.40 | 54.70 | 61.55 | 47.70 
 – |
| *HF Baselines* | | | | | | | | | | | | | | | |
| GTP-2 [9] | 61.75 | 49.90 | 13.90 | 30.55 | 51.70 | 6.50 | 11.7 | 52.10 | 33.10 | 67.60 | 57.50 | 63.60 | 56.10 | 61.50 | 46.60 
 37.40 |
| GPT-BERT (causal) [10] | **67.45** | 49.50 | 34.60 | 36.05 | 52.80 | **6.70** | -3.90 | **68.10** | 46.90 | 74.50 | **68.30** | 76.70 | 56.10 | 65.40 | **54.10** 
 41.20 |
| GPT-BERT (focus-causal) [11] | 62.35 | 49.5 | 31.10 | 32.70 | **52.90** | 6.50 | 3.8 | 67.60 | 51.80 | **78.90** | 67.40 | **77.40** | 57.60 | 61.50 | 53.65 
 40.00 |
| GPT-BERT (mixed-causal) [12] | 65.60 | 50.20 | 25.40 | **48.50** | 25.00 | 6.40 | 14.50 | 66.70 | **53.30** | 77.50 | 67.00 | 76.60 | 55.40 | 63.50 | 52.40 
 39.20 |

Table 1: Evaluation scores on BabyLM tasks for our models (top) and Hugging Face baseline models (bottom). Two macro averages are reported: the first excludes the AoA result obtained from the Hugging Face leaderboard, while the second represents the overall text-average. In table, **BLiMP** refers to the average over BLiMP and BLiMP-supplement, **WUG** corresponds to the average of Wug Adjacency and Wug Past Tense, and **Readings** is the average of Eye Tracking and Self-Paced Reading tasks.

By contrast, **MoEP-SwiGLU** did not reach the same level of performance. This suggests that lightweight linear experts are more effective at the small scale, whereas **SwiGLU** based feed-forward experts require longer training to stabilize and still achieve lower overall scores compared to the other models.

Note that our **GPT-2** and **MoEP-SwiGLU** results do not include **AoA** scores, which are provided in the official **BabyLM** leaderboard.

## 6 CONCLUSION AND DISCUSSION

Despite promising results, **MoEP** and **MoEP-SwiGLU** were trained only on a small dataset and under strict computation budget. It therefore remains unclear whether scaling up the model size and training data would preserve MoEP relative performance compared to other Baseline models. Within **BabyLM**, where the training corpus and patterns to be learned were relatively simple, smaller-dimensional parallel blocks can capture these patterns as effectively as dense layers. With more complex data, however, parallel layers may no longer operate as effectively at reduced dimensionality, forcing an increase in total parameters that could exceed those required by a dense models to learn the same patterns.

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

## A APPENDIX

### A.1 MODEL HYPERPARAMETRS

Comparison of architectural hyperparameters across model variants.

### A.2 TRAINING SETUP

Detailed training configurations for all models.

### A.3 ANALYSIS OF TRAINING DEVELOPMENT

To better understand how each model architecture learns over training, we analyze their fast-evaluation scores across checkpoints. In the following training dynamics analysis, **BLiMP** refers to the average over BLiMP and BLiMP-supplement, **WUG** corresponds to the average of Wug Ad-jacency and Wug Past Tense, and **Readings** is the average of Eye Tracking and Self-Paced Reading tasks.

#### MOEP

Figure 3 presents results for the **MoEP** model. Compared to **GPT-2** (see Figure 4), **MoEP** exhibits more comprehensive early learning, reaching peak performance at the 30M checkpoint, where nearly all task scores are at or above their task-specific means. After 90M words, deviations regress toward zero, with **Entity Tracking** in particular stabilizing well below the mean. This indicates that **MoEP** quickly learns to achieve near-optimal evaluation performance but later begins to overfit, leading

| Hyperparameter | GPT-2 | MoEP | MoEP SwiGLU |
|---|---|---|---|
| Vocabulary size | $\sim$ 16K | $\sim$ 16K | $\sim$ 16K |
| $d_{\mathrm{model}}$ | 384 | 384 / 192 | 384 / 192 |
| Layers | 12 | 2 / 10 | 2 / 10 |
| Parallel blocks | - | 4 | 4 |
| Heads | 6 | 6 / 3 | 6 / 3 |
| Head dimension | 64 | 64 | 64 |
| FF multiplier | 4 | 4 | 4 |
| FF type | MLP | MLP | SwiGLU |
| MoE FF type | - | Liner | SwiGLU |
| N experts | - | 4 | 4 |
| Top k | - | 2 | 2 |
| Normalization | LN | LN | LN |
| Attention | MHA | MHA | MHA |
| Train seq len | 512 | 512 | 512 |
| Total Parameter (millions) | 28M | 28M | 38M |

Table 2: Architectural hyperparameters of GPT-2, MoEP, and MoEP-SwiGLU.

| Hyperparameter | Value |
|---|---|
| Optimizer | AdamW |
| Learning rate | $3 \times 10^{-4}$ |
| Batch size | 16 |
| Training epochs | 10 |
| Gradient accumulation steps | 1 |
| Weight decay | 0.1 |
| Adam betas | (0.9, 0.95) |
| Adam epsilon | $1 \times 10^{-8}$ |
| Scheduler type | Cosine |
| Warmup steps | 800 |
| Random seed | 42 |

Table 3: Training setup and optimization parameters.

to diminished generalization. The pattern highlights that modular routing accelerates initial pattern discovery but may not sustain improvements throughout training.

GPT-2

Figure 4 shows the **GPT-2** baseline smoothed task-mean fast-evaluation results. Unlike **MoEP**, once **GPT-2** reaches its best performance at the 30M checkpoint, it does not stabilize as quickly but continues to improve on certain tasks. On the other hand, after the 70M checkpoint, **WUG** begins to decline and shows no clear signs of stabilization. This reflects a key tradeoff of dense architectures: the model reaches its best scores on different evaluation tasks at different checkpoints rather than converging to a consistent stable state.

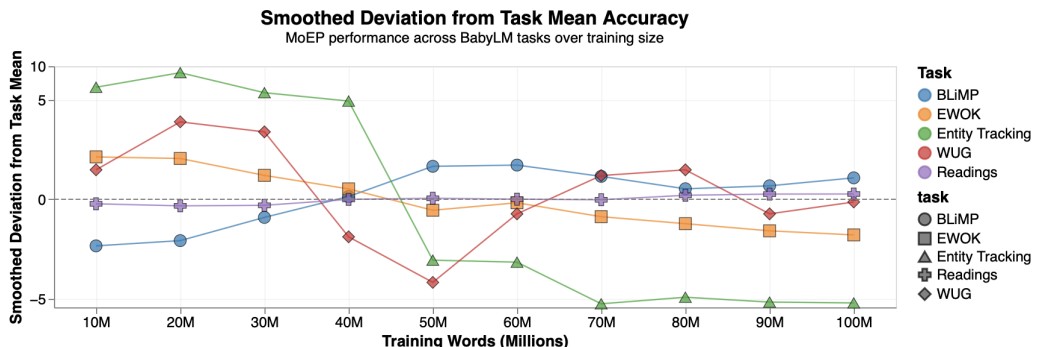

Figure 3: Smoothed deviation from task mean accuracy for MoEP.

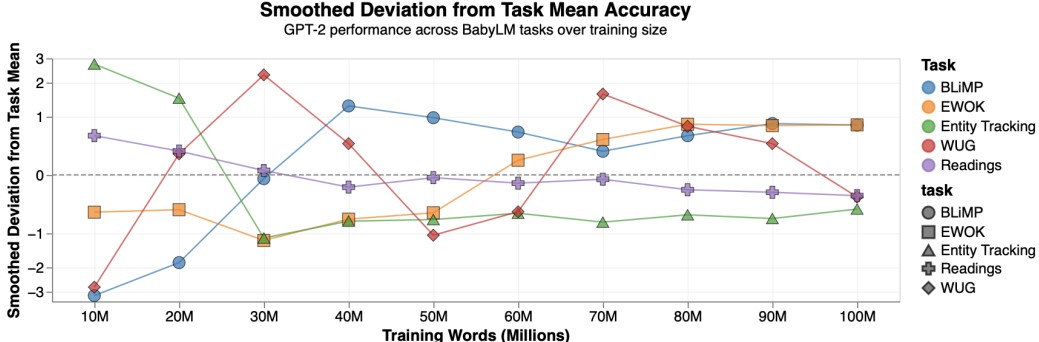

Figure 4: Smoothed deviation from task mean accuracy for GPT-2.

MOEP-SWIGLU

Unlike **MoEP**, **MoEP-SwiGLU** (see Figure 5) shows a development more similar to **GPT-2**. The model exhibits strong late-phase improvements on **WUG** and **BLiMP**, with performance rising steadily after 60M words, while other tasks begin to stabilize. **MoEP-SwiGLU** reaches its best performance at the 80M checkpoint, much later than the other models. As with **MoEP** and **GPT-2**, **Entity Tracking** shows strong instability, where early gains at the first checkpoints collapse sharply afterward. These results suggest that SwiGLU-based experts can improve performance on certain tasks, while other evaluations stabilize without the declines observed elsewhere.

COMPARATIVE TRENDS

**MoEP** learns rapidly during its early checkpoints, showing early specialization particularly on **Entity Tracking** and **WUG**. After the peak at the 30M checkpoint, however, subsequent checkpoints achieve notably lower evaluation scores.

**MoEP-SwiGLU** achieves the strongest late-phase gains on **WUG**, but this comes at the cost of weaker performance on **Entity Tracking** and **BLiMP**.

**GPT-2** shows steadier learning and after reaching its peak performance it experiences fewer dramatic changes in later checkpoints compared to the MoEP variants.

In conclusion, these metrics illustrate that sparse modular routing can accelerate early learning but also introduces instability. The choice of expert type (**linear** vs. **SwiGLU**) further shifts the balance between stability and specialization.

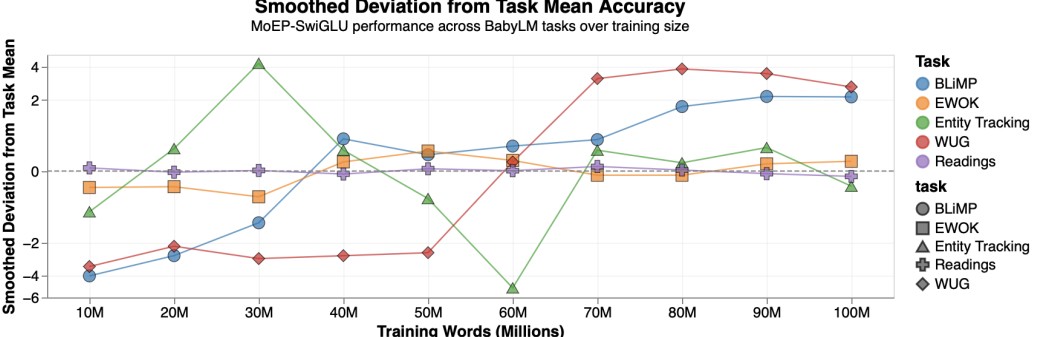

Figure 5: Smoothed deviation from task mean accuracy for MoEP-SwiGLU.

