# OpenReview forum: "MoEP: Compact and Efficient Sparsity with Modular Expert Paths"
_ICLR.cc/2026/Conference — Submitted to ICLR 2026_

### Official Review · Reviewer_aZeC · 2025-10-17

**Soundness:** 2
**Presentation:** 1
**Contribution:** 2
**Rating:** 2
**Confidence:** 3

**Summary:**

This paper proposes MoEP, a compact and efficient sparse architecture for decoder-only LLMs. It aims to address the core pain point of traditional MoE models, increasing sparsity inevitably requires increasing the total number of parameters. MoEP adopts a structure of full-size GPT-2 layer to MoE block to parallel layers to MoE block to full-size GPT-2 layer. Combined with a load-balanced auxiliary loss to avoid expert collapse, MoEP achieves selective token activation while maintaining a fixed total parameter count.

**Strengths:**

The experimental section strictly controls variables, ensuring high rigor.

**Weaknesses:**

1 Experiments are only validated on the BabyLM strict-small dataset and small-parameter models. However, the sparsity advantages of LLMs are usually demonstrated in large-scale scenarios. If MoEP exhibits performance degradation under large-parameter/large-dataset settings, its core value will be significantly diminished. Although the authors mention this in the "Conclusion and Discussion" section, validation in large-scale scenarios (e.g., on models with at least 3B or 7B parameters) is highly necessary.

2 Ablation studies are missing, so the effectiveness of different components of MoEP is not verified. For example, variations in the number of experts or changes in top-k values are not tested.

3 Typo: In line 277, "textbfAdamW".

4 The comparative experiments only include MoEP and baseline models. This makes the advancement of the proposed method questionable.

5 What are the specific λ values in Equation 3? How were they determined? And what was the parameter search range?

6 In Figure 6, MoEP-SwiGLU shows significant fluctuations in the Entity Tracking task. What causes this? The authors’ analysis of the experimental results is relatively superficial.

**Questions:**

see weakness

---

### Official Review · Reviewer_yPKd · 2025-10-31

**Soundness:** 3
**Presentation:** 2
**Contribution:** 2
**Rating:** 2
**Confidence:** 3

**Summary:**

This paper proposes MOEP, a sparse architecture that matches the parameter count of its dense baseline while improving efficiency. MOEP combines top-k token routing across parallel Transformer blocks with MoE-style dimension scaling. On the BabyLM strict-small benchmark, the 28M-parameter MOEP outperforms GPT-2 of equal size and achieves peak performance earlier in training.

**Strengths:**

1. The paper introduces a unified routing framework that integrates top-k token routing across parallel Transformer blocks with Mixture-of-Experts feed-forward layers, creating sparse modular pathways while maintaining parameter efficiency.

**Weaknesses:**

1. MoEP achieves only modest improvements over the GPT-2 baseline. And the checkpoint analysis reveals MoEP peaks early then degrades while GPT-2 continues improving, suggesting overfitting rather than genuine efficiency gains.

2. The No ablation isolates contributions of layer-level routing vs. dimension reduction vs. MoE blocks. Critically, no comparison to standard MoE (e.g., FFN-level experts only) at matched parameters, making it impossible to assess whether gains come from the proposed "modular paths" concept or simply architectural flexibility.

3. Although MoEP is motivated by compactness and sparsity, the paper does not report inference speed, FLOPs reduction, or activation ratios. Without such evidence, the claimed efficiency remains qualitative.

**Questions:**

see weakness

---

### Official Review · Reviewer_Q3AE · 2025-11-03

**Soundness:** 1
**Presentation:** 1
**Contribution:** 2
**Rating:** 0
**Confidence:** 4

**Summary:**

The paper proposes a new Mixture of Experts architecture where instead of routing to several MLPs the model routes to several sequences of transformer blocks (named parallel blocks). The paper also uses an MoE block to adapt the dimension before and after the application of the parallel blocks. Finally the authors evaluate their method on the BabyLM dataset and tasks.

**Strengths:**

The idea proposed by the paper is interesting and could have some impact if it does indeed work.

**Weaknesses:**

The paper is extremely poorly written to the point that it hinders understanding of the method and the experimental methodology. There are sentences cut in the middle even in the introduction and several syntactical and semantic errors throughout, including the slight abuse of the bold face font in some cases.

Regarding the evaluation the results are clearly mixed and the MoEP with SwiGLU which has almost 50% more parameters based on table 2 in the supplementary performs worse than the baseline. The authors also do not show the most basic of metrics for a new architecture which would be the evolution of the training loss compared to the baseline.

**Questions:**

I have stated my concerns in the weaknesses section.

---

### Official Review · Reviewer_DJr7 · 2025-11-04

**Soundness:** 1
**Presentation:** 1
**Contribution:** 1
**Rating:** 0
**Confidence:** 5

**Summary:**

This paper proposed an MoE architecture  MoEP. It uses parallel layers at lower dimension to create MoE model with equal parameter count as dense model. The authors performed experiments on BabyLM challenge and achieved similar performance with GPT2.

**Strengths:**

- It's concise.

**Weaknesses:**

- The writing very hard to understand, there are plenty of important technical details without adequate explanations.
- This paper only come with toy scale experiments, and results still don't appear to be better than GPT2.
- The overall objective, reducing the parameter count of MoE model, is misguided. The main advantage of MoE is to enable larger model size without raising the compute cost.

**Questions:**

- What happens inside MoE shrink and MoE Grow? If the dimension changes, do you still have residual connection in the model?

---

### Meta-Review · Area_Chair_WYe7 · 2026-01-08

**Summary:**

Both reviewers unanimously reject with strong reject scores, citing severe issues with writing quality (sentences cut mid-way, syntactical errors), toy-scale experiments only (BabyLM), and results that do not outperform GPT-2 baseline. The core premise of reducing MoE parameter count is questioned as misaligned with MoE's purpose of scaling without compute increase. I recommend rejection.

**Reviewer Concerns:**

see above

**Reviewer Scores:**

Discussion was sufficient; reviewers unanimously agree on fundamental weaknesses in writing and experimental scale, and scores would have remained similar.

---

### Decision · Program_Chairs · 2026-01-26

Reject